# Iodine and Mercury Content in Raw, Boiled, Pan-Fried, and Oven-Baked Atlantic Cod (*Gadus morhua*)

**DOI:** 10.3390/foods9111652

**Published:** 2020-11-12

**Authors:** Lisbeth Dahl, Arne Duinker, Synnøve Næss, Maria Wik Markhus, Ive Nerhus, Lisa Kolden Midtbø, Marian Kjellevold

**Affiliations:** Department of Seafood and Nutrition, Institute of Marine Research (IMR), NO-5817 Bergen, Norway; Arne.Duinker@hi.no (A.D.); synnoeve.naess@hi.no (S.N.); Maria.Wik.Markhus@hi.no (M.W.M.); i.nerhus@gmail.com (I.N.); Lisa.Kolden.Midtbo@moreforsk.no (L.K.M.); Marian.Kjellevold@hi.no (M.K.)

**Keywords:** fish, cooking, processing, food composition data, analysis, ICP-MS, DMA-80, nutrition security

## Abstract

There is a lack of scientific evidence regarding the stability of iodine and mercury during cooking and processing of seafood. In this study, the iodine and mercury content were determined after thawing frozen fillets of Atlantic cod (*Cadus morhua*), and further in raw compared to boiled, pan-fried, and oven baked fillets. Iodine was determined by Inductively Coupled Plasma-Mass Spectrometry (ICP-MS) and mercury by atomic absorption spectrophotometry with Direct Mercury Analyzer (DMA-80). Thawing of the cod resulted on average in a 12% loss of iodine to the thawing water. Boiling significantly decreased the total content of iodine per slice of cod fillet corresponding to the concentration of iodine found in the boiling water. Pan-frying and oven-baking did not cause any significant changes of the total iodine content per slice of cod fillet, although iodine content per 100 g increased due to weight reduction of the cod slices from evaporation of water during preparation. For mercury, we found minimal changes of the different cooking methods. In summary, the findings in our study show that boiling had the greatest effect on the iodine content in the cod fillets. Type of cooking method should be specified in food composition databases as this in turn may influence estimation of iodine intake.

## 1. Introduction

Representative, local, up-to date high-quality food composition data are of fundamental importance for estimating nutrient intake in a population. Chemical analysis is the gold standard for generating data, but due to costs, analytical data comprise only a minor part of available information in most food composition databases (FCDB) and in food composition tables (FCT) [1]. Changes in nutrient content due to cooking method; for instance, boiling, frying, and grilling, and processing such as smoking, salting, and drying, should be considered when estimating nutrient intake of different foods [2]. Storage of fish by freezing is an often-used method to increase the shelf life of seafood. Thus, any changes of nutrient content during thawing, and further use of processing and cooking methods of the fish is important to quantify. 

Presently, the Norwegian FCT [3] and the International Network of Food Data Systems (INFOODS) Food composition database (FCDB) [4] for fish and shellfish [5] only include analytical data on iodine in raw fish. In both the INFOODS FCDB [4] and the Norwegian FCT [3], values for cooked, fried, and processed fish are estimated or calculated, not chemically analyzed. Analytical data on contaminants and heavy metals are not included in the Norwegian FCT, but such data in seafood are available in the open access Seafood database [6] at Institute of Marine Research (IMR). Atlantic cod (*Gadus morhua*) is the most commonly consumed lean white fish species in Norway, and total catch was 327,648 tons (live weight) in the Norwegian fisheries in 2019 [7]. Raw Atlantic cod and other lean fish species such as pollack (*Pollachius pollachius*), haddock (*Melanogrammus aeglefinus*), and saithe (*Pollachius virens*) have a considerably higher content of iodine than fatty fish species like mackerel (*Scomber scombrus*), herring (*Clupea harengus*), and farmed Atlantic salmon (*Salmo salar*) [8,9]. Iodine is an essential micronutrient for humans and has an important role in growth, brain development, and metabolism as it is active in the biosynthesis of thyroid hormones [10]. At the same time, the range for acceptable iodine intake for adults is from 150 to 600 μg/day and is considered relatively constricted compared to other micronutrients [11]. Thus, it is important to have high quality data on iodine in iodine rich foods such as Atlantic cod. However, Atlantic cod is also a source of mercury and methylmercury. This is of concern in relation to food safety, as humans are predominantly exposed to mercury through fish consumption [12]. The content of mercury in different fish species varies considerably and depends on factors such as type of species, geographical area, size, and age [13]. Knowledge about mercury exposure in humans is of importance as mercury in the form of methylmercury may have adverse effects such as impaired neurodevelopment of the fetus during pregnancy [14]. We have previously published analytical data on iodine and mercury in several raw lean fish species [13,15,16] and in processed fish products [17,18]. In the present study, the main aim was to analyze the total iodine and mercury content in Atlantic cod fillets after thawing and further after three different cooking methods, i.e., boiling, pan-frying, and oven-baking. To our knowledge, this is the first experiment presenting analytical data on iodine and mercury content in Atlantic cod during different cooking methods.

## 2. Materials and Methods

### 2.1. Fish Samples

The study included Atlantic cod (*Gadus morhua*) catched in the Barents Sea in October 2015 purchased from Lerøy Seafood Group ASA (bought after tender). The cod weight ranged from 1–2.5 kg and was immediately frozen in blocks of 25 kg, and stored as whole fish without the head and organs at −30 ℃. In December 2015, the cod was thawed and fillet portions of approximately 200 g without skin were produced and frozen separately in strings of four fillets in each package (Bulandet Fiskeindustri AS, Lerøy Seafood Group ASA) before transported frozen to the IMR and stored in an outdoor freezing room at −30 ℃ pending analysis and delivery to participants in the randomized intervention study “Mommy food” [19].

### 2.2. Practical Procedure and Sample Preparation

The cod fillet packages were thawed overnight for approximately 18 h in a refrigerator at 4 °C before the experiment was performed in July 2017. In total, 30 cod fillets were used in the present study. Each fillet was divided into two portions of approximately 100 g each. Half of each cod portion of 100 g was kept raw (*n* = 30). The other half was further prepared for three different cooking methods either 1: boiled (*n* = 10), 2: pan-fried (*n* = 10), or 3: oven-baked (*n* = 10). All cod fillets were weighted before and after processing. Figure 1 shows a schematic overview of the sample preparation of the cod fillets.

For cooking method 1, boiling, each cod fillet of approximately 100 g was put into a pan of boiling water and then stirred for 10 min in 1 L of water. For cooking method 2, pan-frying, the cod fillets were pan-fried for 6–7 minutes in 10 mL of rapeseed oil at medium temperature. For cooking method 3, oven-baking, the cod fillets were baked in the oven at 180 °C for 15 min. Thawing- and boiling water and any liquid left after pan-frying or oven baking were also collected for analyses of iodine. The experiment was performed using a household ceramic electric cooker (Gorenje, SuperPower Induction), a stainless-steel cooking pan (3 L of size), aluminum frying pan covered with Teflon (diameter 28 cm), and disposable aluminum form (0.5 L of size). No salt, spices, or food additives were used in the different cooking procedures.

After the experiment, all cod samples were freeze-dried (Labconco Freezone 18L Mod.775030, Kansas City, MI, USA) to constant weight using an accredited method according to ISO 17025. The samples were homogenized, and a sub-sample of the wet sample were weighed individually, put in separate plastic containers and frozen at minus 20 °C overnight. The samples were freeze-dried for 72 h (24 h at −50 °C, immediately followed by 48 h at +25 °C, with a vacuum of 0.2–0.01 mbar). The samples were then weighed once again, and the dry matter was calculated based on the difference in weight of the sample before and after freeze-drying. The method is validated, and measurement uncertainty is 10% for dry weight samples in the range of >10 to 99.5 g/100 g. Freeze-dried samples were then homogenized to a fine powder using a domestic mill and stored in twist off boxes at room temperature pending analysis. In total, 60 samples of cod (raw (*n* = 30) and processed (*n* = 30)) and 60 samples of liquid (i.e., thawing water (*n* = 30), boiling water (*n* = 10), liquid after pan-frying (*n* = 10), and oven baking (*n* = 10)) were collected. Iodine was determined in all samples (*n* = 120) and mercury was determined only in the cod fillet samples (*n* = 60), i.e., not in the liquid samples. To evaluate if the different cooking methods had any effect on iodine and mercury, we assessed the iodine and mercury content (µg) per piece of cod fillet and per 100 g (µg/100 g) cod fillet before and after cooking.

### 2.3. Determination of Iodine

For the determination of iodine, subsamples of ~0.2 g dry weight were added to 1 mL ultrapure tetrametylammonium hydroxide (TMAH) and 5 mL deionized water (>17 MΩ cm^−1^, Nano pure-system, Nanopure, Barnstead, UK) before extraction at 90 °C ± 3 °C for 3 h. For the extraction of the liquid samples, 4 mL of liquid was added to 1 mL TMAH. The liquid- and the cod samples were, after extraction, diluted to 10 mL and 25 mL, respectively with deionized water and left overnight for sedimentation of any solid particles. Aliquots of 10 mL were pipetted from the middle of the tubes in order to avoid taking up any precipitate from the bottom part of the solution. Prior to quantification, the samples were filtered through a 0.45 µm single use syringe and disposal filter. The 1% TMAH solution contained tellurium (1 mg/L) which was used as an internal standard in order to correct for instrument drift. Samples were analyzed against a standard addition calibration curve (2, 5, 10, 20, and 50 µg/L) to measure the unknown iodine content in the samples. Iodine was determined by Inductively Coupled Plasma-Mass Spectrometry (ICP-MS) with an iCap Q ICP-MS (Termo Fisher Scientific, Waltham, MA, USA) equipped with an autosampler (FAST SC-4Q DX, Elemental Scientific, Omaha, NE, USA). Limit of quantification (LOQ) is 0.32 µg/L or 0.04 mg/kg dry weight. Limit of detection (LOD) is 0.01 μg/L. The measurement uncertainty differs depending on the concentration range and is set to 15% for concentrations >10 × LOQ and 40% for concentrations between LOQ and 10 × LOQ. The trueness of the method was evaluated by analysis of Certified Reference Material (CRM). The CRM value for Fish muscle (BB 422) and skim milk powder (ERM-BD 150) is 1.4 ± 0.40 mg/kg and 1.73 ± 0.14 mg/kg for iodine, receptively. The trueness for iodine in CRMs used in the present study (*n* = 6) was in good agreement with the certified values and with the control chart of these two CRMs.

### 2.4. Determination of Mercury

For the determination of mercury, the cod samples were analyzed for total mercury by thermal decomposition, amalgamation, and atomic absorption spectrophotometry [20] using a Direct Mercury Analyzer (DMA-80, Milestone Srl, Italy). DMA-80 is calibrated in the linear area of mercury from 1.5–1000 ng. For samples in this area the accuracy is 80–120%. Samples were weighed on a calibrated four decimal scale from Sartorius (CP124S, Goettingen, Germany) and positioned in separated nickel boats prior to analyses. There are 40 positions for metal boats per analysis series in DMA-80. For each analysis series, there were empty metals boats at position 1 and 2 to make sure of no contamination from previous analyses. TORT-3: Lobster Hepatopancreas Reference Material for Trace Metals was used as CRM. A total of six samples of TORT-3 were placed at the beginning (*n* = 2), middle (*n* = 2), and end (*n* = 2) of the analysis series to check the accuracy of the method throughout the analysis. The CRM value for TORT-3 is 292 µg/kg for total mercury. Mean ± SD of analyzed CRM (*n* = 6) was 295 ± 11.4 µg/kg for total mercury, giving a mean accuracy of 101% (% relative SD: 3.9%). All results were within the accepted area of the analyses (±20%). Mercury determination was performed in two consecutive series and all analyzed values were above the LOQ of 0.08 ng mercury and the LOD of 0.02 ng.

### 2.5. Data Analysis and Statistical Methods

The descriptive statistics mean, standard deviation (SD) of mean, median, and range were conducted using Microsoft Office Excel 365 ProPlus (Microsoft Corporation, Redmond, WA, USA). For testing changes during cooking, percentage change from before to after cooking was used as this also normalized some of the variation from a few samples with high iodine levels. Statistica 13 (©Statsoft, Tulsa, OK, USA) was used to test if cooking methods had impact on the iodine and mercury content as well as dry weight percentages, using t-tests on percentage change, to test if these were different from zero.

## 3. Results

### 3.1. Total Iodine and Mercury Content in Raw Cod Slices after Thawing

The iodine and mercury content in thawed raw cod (*n* = 30) (~100 g/piece of cod fillet) and iodine in the associated thawing water are shown in Table 1, given as total µg per piece of cod fillets or in thawing water. The mean ± SD and median iodine content in the raw thawed cod fillets (*n* = 30) were 72 ± 87 µg and 52 µg wet weight, respectively. The iodine content (µg) ranged from 29.8 µg (cod #22) to 512.8 µg (cod #10) in the raw thawed cod fillets. Relative loss of iodine into the thawing water ranged from 5.5% (cod #23) to 19.9% (cod #21) with an average loss of 11.6 ± 3.4% (Table 1). The sample with the highest iodine content (cod #10) had similar loss into the thawing water compared to the other samples with lower iodine content. The mercury content in the raw thawed cod ranged from 1.3 µg (cod #19) to 6.3 µg (cod #13) per cod fillet wet weight with a mean ± SD and median mercury content of 2.6 ± 1.4 µg and 2.2 µg wet weight per cod sample (*n* = 30), respectively.

### 3.2. Dry Weight and Weight of the Cod Fillets before and after Different Cooking Methods

The dry weight (%) and weight (g) of the cod fillets before and after the different cooking methods are reported in Table 2. The mean ± SD dry weight in the cod fillets (*n* = 30) was 19.1 ± 0.6% before and 23.3 ± 2.0% after the different cooking methods. The dry weight increased significantly (*p* < 0.01), with the highest increase after pan-frying of the fillets.

The weight of the raw cod fillet (*n* = 30) varied from 77.2 g (cod #11) to 110.1 g (cod #23) with a mean ± SD weight of 101.3 ± 7.2 g. The weight decreased significantly (*p* < 0.01) in all fillets after the different cooking methods and the weight loss ranged from 9.1% to 24.3% with a mean ± SD and median of 18.2 ± 3.3% and 18.6%, respectively.

### 3.3. Iodine and Mercury Content per Cod Fillet and Content per 100 g before and after Different Cooking Methods

The iodine content per fillet (µg) and per 100 g fillet (µg/100 g) before and after boiling, pan-frying, and oven-baking the cod fillets are shown in Table 3. Boiling the cod fillets (*n* = 10) in one liter of water reduced the iodine content per fillet with approximately 20%, but was not significant. The mean ± SD iodine concentration in the boiling water was 32 ± 43 µg and corresponded to an approximately 30% loss of iodine to the boiling water. The iodine content given as per 100 g fillet was significantly decreased by approximately 10%. Pan-frying and oven-baking did not cause any significant changes to the total iodine per fillet, although iodine content per 100 g increased due to weight reduction of the cod slices from evaporation of water during preparation. Figure 2 shows the percentage change of iodine given as total iodine content per fillet (µg) and as iodine content per 100 g (µg/100 g) before and after the different cooking methods.

The mercury content before and after boiled (*n* = 10), pan-fried (*n* = 10), and oven-baked (*n* = 10) cod samples are shown in Table 4. The mean ± SD mercury per cod fillet in all cod samples (*n* = 30) was 2.8 ± 1.5 µg before and 2.8 ± 157 µg wet weight after the different cooking methods and were not significantly different from each other (*p* > 0.05). The mercury content per 100 g fillet (µg/100 g) increased after the three different cooking methods using wet weight. Since mercury will not dissolve in the water phase, we used the dry weight data and found no significant (*p* > 0.05) differences between the different cooking methods.

## 4. Discussion

The present study has assessed the effect of thawing, boiling, pan-frying, and oven-baking on the content of iodine and mercury in Atlantic cod fillets. In general, thawing of frozen cod fillets caused an approximately 12% loss of iodine to the thawing water. Boiling significantly decreased the total iodine per piece of cod fillet and the corresponding amount of iodine was found in the boiling water. Pan-frying and oven-baking did not cause any significant changes in the iodine per fillet, although iodine content per 100 g increased due to weight loss of the slices from evaporation of water during cooking. For mercury, we found minimal changes of the different cooking methods.

We found that boiling of the cod fillets significantly decreased the total content of iodine per piece of cod and that this loss was almost equal to the iodine concentration found in the boiling water. Thus, the average iodine loss in µg in the cod fillets can be explained by loss to the boiling water. The iodine content per 100 g fillet was reduced due to increased dry weight and reduced weight of the cod fillet after boiling. Steaming (105 °C in aluminum foil for 15 min) has previously been shown to not affect iodine content in hake (*Merlucius australis*), monkfish (*Lophius piscatorius*), mackerel (*Scomber scombrus*), tuna (*Katsuonus pelamis*), plaice (*Pleuronectes platessa*), mussel (*Mytilus edulis*), octopus (*Octopus vulgaris*), and shrimp (*Litopenaeus vannamel*) [21]. In the study by Doh et al., 2019, they found more than 60% loss of iodine after boiling abalone (*Haliotis discus hannai*) and 32% reduction after steaming or grilling [22]. Some of the explanation for higher loss of iodine from abalone may be that they are invertebrates, with open circulatory system, which may lead to greater loss of water-soluble components during boiling. Invertebrates also have higher concentrations of osmolytes in the tissues as they are osmoconformers and this may further contribute to greater loss of iodine. Although the loss is different between our study and Doh et al., 2019, it is reasonable that some iodine will be lost during boiling as iodide is reactive with the potential to undergo oxidation and reduction reactions within the food matrix [22]. Even if the iodine loss after boiling was higher per fillet than per 100 g cod fillet in our study, our results indicate a reduction of iodine in the range of 10–20%.

Most studies have primarily assessed the fate of iodized salt in a variety of foods after different cooking methods [23,24,25,26], and there are few studies investigating the effect different cooking- and processing methods may have on iodine content in fish and other seafood. In the study by Longvah et al., 2013, they reported an average loss of 47% iodine in different recipes from the Indian kitchen using iodized salt after boiling and the range of loss was from 14 to 88% [24]. The same study also reported a minor loss of iodine with cooking methods such as steaming, deep frying, and pressure cooking of the different recipes. Loss of iodine from the use of added iodized salt in food/recipes is not exactly comparable to the aim in the present study, but will be relevant when estimating iodine intake from the diet.

Pan-frying and oven-baking of the cod fillets showed minor changes of the total content of iodine (µg) per piece of cod in the present study. As a result that the dry weight percentage increased and the weight of the cod fillets decreased due to evaporation of water, we observed that the iodine content per 100 g (µg/100 g) in wet weight increased with 15–20% after pan frying or oven baking. We therefore assume that these two cooking methods are better in regard to preserving the iodine content compared to boiling. However, if cod is part of a dish where also the water is consumed (e.g., soup), the iodine loss will be less and comparable to pan-frying and oven-baking.

Different studies investigating mercury in seafood shows that cooking in general tends to increase the wet weight content of mercury in seafood, most likely due to loss of moisture during the cooking process [12,27,28]. Our results support these findings regarding mercury since we also observed a minor decrease of the moisture after pan-frying and oven-baking. In a study with Spanish mackerel (*Scomberomorus maculatus*), cat shark (*Scyliorhinus sp.*), and red tuna (*Thunus* thynnus), the dry weight mercury content for all these three fish species were slightly higher after boiling compared to frying and raw fish [29]. In the same study, they found that boiling and frying reduced mercury bioaccessibility by 40% and 60%, respectively, compared to raw fish mercury bioaccessibility. Bioaccessibility is the proportion of the mercury that potentially reaches the systemic circulation. The mercury bioaccessibility ranged from 10% in octopus (*Octopus vulgaris*) to 60% in monkfish (*Lophius piscatorius*) [21]. Thus, although the mercury after different cooking methods is almost unchanged in the present study, the estimation of exposure of mercury can probably be overestimated due to a reduced bioaccessibility of the mercury after cooking [21]. However, this was not assessed in this experiment and must be explored further.

As frozen cod were used in the present study, we were able to study if there were any changes of iodine content during thawing. The cod was frozen and thawed twice before we performed the different cooking methods of the portion packed cod. Given an average 12% loss of iodine due to thawing, the fresh caught cod may therefore originally have had up to a 25% higher iodine concentration. Using this approach, the iodine content in the fresh caught cod was approximately 100 µg/100 g fillet. However, this finding has probably no important relevance for estimating iodine intake from cod, since cod should not be eaten raw due to parasites.

The large variation of iodine between and within fish species, but also in relation to condition factor (100 × weight/lenght^3^), season, and geographical location [9,15], is a challenge when estimating the iodine intake from cod. The reported range of iodine in raw Atlantic cod fillets was 22 to 720 µg/100 g (*n* = 121) in the paper by Nerhus et al., 2018 and 18 to 1270 µg/100 g (*n* = 125) in the paper by Julshamn et al., 2001 [8]. Atlantic cod is regarded as a fish species relatively low in mercury; however, there are also large intraspecies variations. Still, the mercury content in cod varies less compared with iodine and ranged from 1 to 54 µg/100g (*n* = 516) with a mean of 11 ± 7 µg/100 g in a study from the North Sea and costal Norwegian waters and from 1 to 16 µg/100g (*n* = 804) with a mean of 3.6 ± 2.3 µg/100 g in samples from the Barents Sea [15]. The mercury content in the present study was approximately 3 µg/100 g raw fillet.

In the Norwegian FCT [3] boiled (fillet), oven-baked, and pan-fried are given the same value as raw cod (279 µg/100 g), while sliced cod has lower value (194 µg/100 g). In the INFOODS FCDB, boiled and grilled cod are given values higher than raw as seen in the present study as well, ranging from 280–400 µg/100 g. In both the INFOODS FCDB [4] and the Norwegian FCT [3], these values are estimated. Our study contributes with novel and new data on the effect of different cooking methods on the iodine level that further can be implemented in FCDBs and FCTs. With the shortage of studies regarding the effect of the cooking process on iodine, we can only speculate and assume that loss of iodine after boiling is most likely the same for other lean fish species.

Representative and reliable analytical food composition data are considered essential for estimating and evaluating the nutrient intake of individuals and population groups. To estimate intake of iodine and mercury exposure, representative and reliable analytical data are essential. Therefore, our study will provide new insights and reliable information that will increase the quality of iodine values after different cooking methods of cod fillets in FCDBs and FCTs, and further the quality of data reporting the dietary iodine intake. FCDBs and FTCs would also benefit in including the most important food safety parameters like e.g., mercury. A limitation of our study is the relatively low number of samples for each cooking method and the smaller variation in the iodine content compared to other studies reporting iodine content.

## 5. Conclusions

The present study has determined the effect of thawing and different cooking methods on the content of iodine and mercury in Atlantic cod. Boiling decreased the iodine content per fillet and per 100 g fillet with approximately 10–20%. Pan-frying and oven-baking caused minor changes in the net iodine content, while content per 100 g increased due to reduced moisture. For mercury, we found minimal changes of the different cooking methods. Further studies are warranted to better understand the variation in iodine content in different fish species fillets and type of processing should be specified in food composition databases.

## Figures and Tables

**Figure 1 foods-09-01652-f001:**
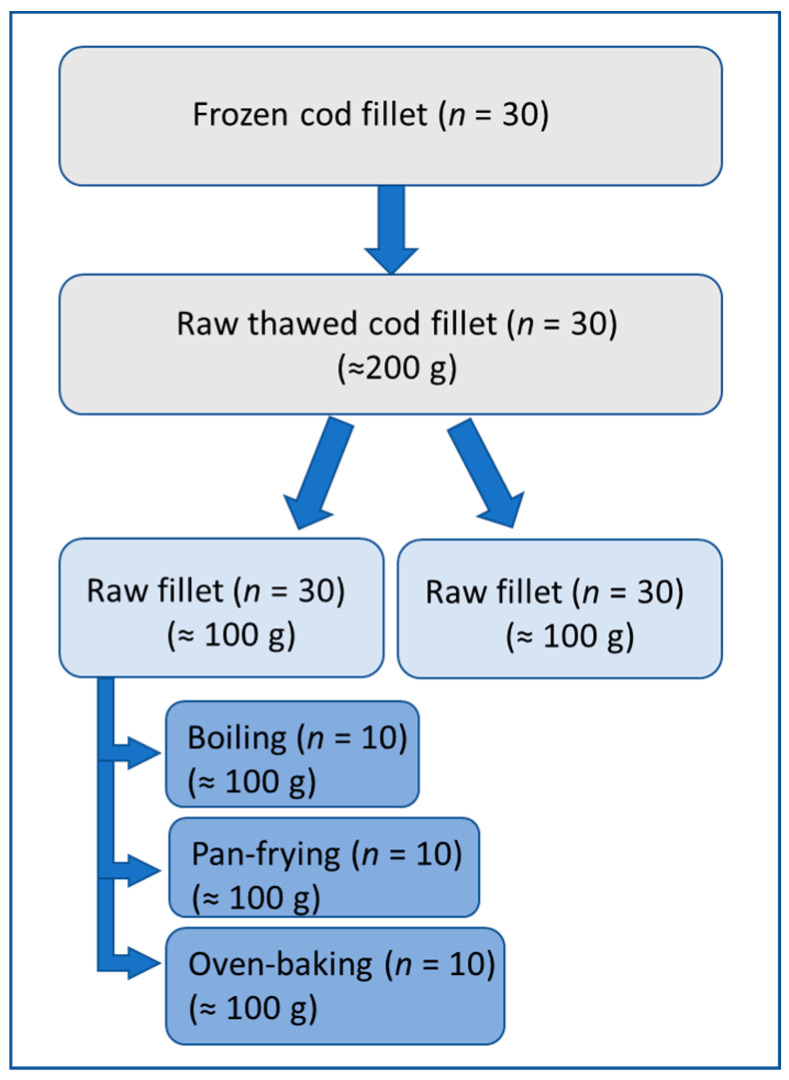
Schematic overview of cod fillet processing.

**Figure 2 foods-09-01652-f002:**
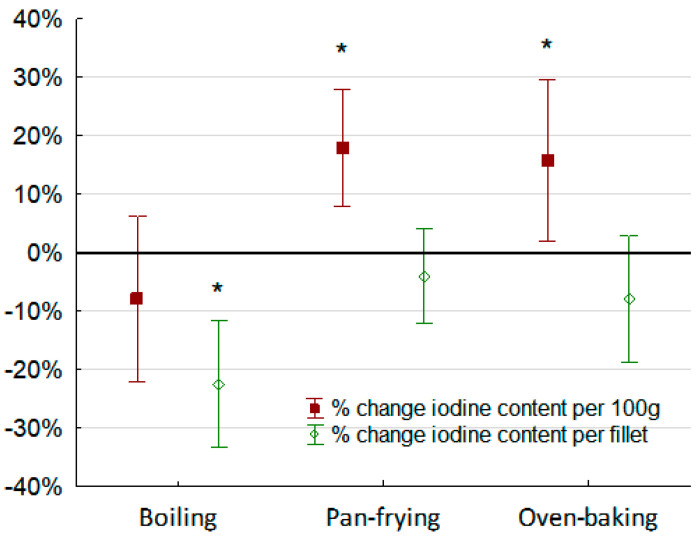
Percentage (%) change of iodine before and after different cooking methods (wet weight). Red bars show change in iodine content (µg/100 g) and green bars show change in total iodine per cod fillet (µg). Significant differences within cooking method are indicated by * (*p* < 0.01).

**Table 1 foods-09-01652-t001:** Total iodine and mercury (µg) in raw cod fillets after thawing and iodine in thawing water (µg and %). Numbers are given as mean ± SD and (median) wet weight.

Sample	Iodine/Raw Fillet * (µg)	Iodine Thawing Water (µg)	Iodine Loss after Thawing (%)	Mercury/Raw Fillet (µg)
Cod fillet (#1–10)	101.4 ± 146.4	13.0 ± 19.6	12.3 ± 1.9	2.7 ± 1.3
	(53.0)	(6.0)	(12.2)	(2.3)
Cod fillet (#11–20)	59.7 ± 32.7	7.2 ± 4.3	12.2 ± 3.1	3.0 ± 1.8
	(50.0)	(6.7)	(13.3)	(2.4)
Cod fillet (#21–30)	54.7 ± 15.5	5.2 ± 1.8	10.2 ± 4.6	2.1 ± 0.8
	(53.4)	(4.8)	(8.9)	(2.0)
All (*n* = 30)	71.9 ± 86.7	8.5 ± 11.7	11.6 ± 3.4	2.6 ± 1.4
	(52.0)	(5.4)	(11.7)	(2.2)

* The other half of the cod fillets #1–10 were boiled, #11–20 were pan-fried, and #21–30 were oven-baked.

**Table 2 foods-09-01652-t002:** Dry weight (%) and weight (g) of the cod fillets before and after the different cooking methods and percent change of weight in wet weight. Numbers are given as mean ± SD and (median).

Cooking Method	Dry Weight before (%)	Dry Weight after (%)	Weight before (g)	Weight after (g)	Weight Change (%)
Boiling (*n* = 10)	18.9 ± 0.6 ^a^	21.0 ± 0.7 ^b^	99.6 ± 7.4 ^a^	84.2 ± 7.5 ^b^	15.5 ± 3.5
	(19.0)	(20.7)	(99.8)	(84.5)	(16.6)
Pan-frying (*n* = 10)	19.0 ± 0.5 ^a^	25.6 ± 0.9 ^b^	100.7 ± 7.1 ^a^	82.1 ± 7.4 ^b^	18.6 ± 1.9
	(19.0)	(25.7)	(100.8)	(81.6)	(18.0)
Oven-baking (*n* = 10)	19.5 ± 0.6 ^a^	23.2 ± 0.5 ^b^	103.6 ± 7.3 ^a^	82.5 ± 7.4 ^b^	20.4 ± 2.4
	(19.4)	(23.2)	(100.2)	(81.8)	(20.0)
All (*n* = 30)	19.1 ± 0.6	23.3 ± 2.0	101.3 ± 7.2	82.9 ± 7.2	18.2 ± 3.3
	(19.0)	(23.2)	(101.2)	(20.0)	(18.6)

Different letters denote significantly differences between dry weight rows and between weight rows (*p* < 0.01).

**Table 3 foods-09-01652-t003:** Total iodine (µg) per piece of cod fillets and in the associated liquid (µg), and iodine content (µg/100 g) in cod fillets before and after the different cooking methods. Numbers are given as mean ± SD and (median) wet weight.

Cooking Method	Iodine/Fillet, before * (µg)	Iodine/Fillet, after ** (µg)	Iodine Liquid *** (µg)	Iodine Content before (µg/100 g)	Iodine Content after (µg/100 g)
Boiling (*n* = 10)	114.6 ± 175.1 ^a^	79.4 ± 101.9 ^b^	32.5 ± 43.4	116.9 ± 182 ^a^	97.6 ± 130.8 ^a^
	(55.3)	(40.5)	(14.8)	(56)	(44.5)
Pan-frying (*n* = 10)	65.9 ± 29.1	63.9 ± 30.4	1.3 ± 0.5	66.7 ± 33.8 ^a^	79.5 ± 43.9 ^b^
	(56.2)	(49.5)	(1.2)	(57)	(62.5)
Oven-baking (*n* = 10)	61.6 ± 18.1	55.7 ± 15.4	9.3 ± 3.4	59.1 ± 15.6 ^a^	67.2 ± 16.4 ^b^
	(62.4)	(57)	(16.6)	(60.5)	(70)
All (*n* = 30)	80.7 ± 102.4	66.3 ± 60.7	14.4 ± 27.8	80.9 ± 106.7	81.4 ± 78.5
	(57.6)	(49.4)	(9.6)	(56.5)	(62.5)

* Iodine was calculated by multiplying the analyzed iodine concentration in the raw cod slice with the weight of the cod slice before cooking. ** Iodine was calculated by multiplying the analyzed iodine concentration after cooking with the weight of the cod slice after cooking. Different letters denote significant differences between rows *p* = 0.001. *** Iodine in liquid (%) was calculated by dividing analyzed iodine concentration in the liquid with analyzed iodine content in the raw cod slice and then dividing by 100. Different letters denote significant differences between iodine per fillet rows and between iodine content per 100 g fillet rows (*p* < 0.01).

**Table 4 foods-09-01652-t004:** Total mercury per fillet (µg) and mercury content per 100 g fillet (µg/100 g) in cod fillets before and after the different cooking methods. Numbers are given as mean ± SD and (median) wet weight.

Cooking Method	Mercury/Fillet, before (µg)	Mercury/Fillet, after (µg)	Mercury Content before (µg/100 g)	Mercury Content after (µg/100 g)
Boiling (*n* = 10)	2.9 ± 1.6	2.7 ± 1.4	2.9 ± 1.3	3.2 ± 1.5
	(2.5)	(2.4)	(2.5)	(2.8)
Pan-frying (*n* = 10)	3.3 ± 1.9	3.5 ± 1.9	3.3 ± 1.8	4.3 ± 2.2
	(2.7)	(3.1)	(2.7)	(3.6)
Oven-baking (*n* = 10)	2.4 ± 0.8	2.2 ± 0.8	2.3 ± 0.8	2.7 ± 1.0
	(2.1)	(2.0)	(2.1)	(2.5)
All (*n* = 30)	2.9 ± 1.5	2.8 ± 1.5	2.8 ± 1.4	3.4 ± 1.7
	(2.4)	(2.3)	(2.4)	(2.8)

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
