# Peer review of "Iodine and Mercury Content in Raw, Boiled, Pan-Fried, and Oven-Baked Atlantic Cod (Gadus morhua)"

_foods, 2020, doi:10.3390/foods9111652_

Round 1
Reviewer 1 Report
The manuscript „Iodine and mercury content in raw, boiled, pan-fried and oven-baked Atlantic cod (Gadhus morhua)“ brings additional, but not the first one, information about changes of iodine and mercury content as consequence of various cooking procedures.
The structure of the manuscript is classic and clear.
The biggest problem of the manuscript are major shortcomings in the methodology:
- Row 104: describe procedure (oven, freeze-drying, ….) and conditions (temperature, time) of drying. Additionally, results are given in wet weight – why if dry samples were extracted?
- Row 104/105: add concentration of TMAH; Really pure TMAH was used? Because it is often in form of pentahydrate.
- Row 109: sedimentation of solid particles, but only liquid was analysed. Has it been verified that the solid particles do not contain any residual iodine? Specify extraction efficiency as ratio of total iodine in extract and total iodine in sample for fish samples, not for CRMs! (Because CRMs are really good disintegrated and homogenized compared to common samples).
- Row 112: add concentration of tellurium
- Row 113: specify number and concentration of standard additions, it is crucial for method precision
- Row 117/118: LOD and LOQ are the same 0.01 ug/l, correct values
- Row 124: n=2; two replicates are really insufficient for evaluation of trueness, n=4 are ok
- Row 151 and Table 1: average mass of iodine per fillet is senseless because mass of all fillets is not 100 g as provide in row 150, but varies from 77 to 110 g (row 168). Data are incomparable with this variability. Why is mass of iodine in fillet 1-10 cca two times higher than in 11-20 and 21-30? Please provide also mercury in thawing water.
- Row 167: significance p<0.001 is given here, but p<0.05 in Table 2 – explain that
- Row 169: weight of raw fillet is 91.1 g here, but 101.3 in Table 2 – correct the value
- Table 3: the information of iodine before (ug) and iodine after (ug) is unnecessary because if the fillet has ca 100 g, the number is similar to contents in (ug/100 g). Please remove this duplicate. The explanation of *** mark is missing. Also “different letters” mention below table are missing. Medians are very low in comparison to average values in some cases, this points to high outliers – please explain that.
- Figure 2: insufficient quality of picture, what the symbol * mean in chart?
- Row 199: the mean of Hg is 2.8 here, but 2.9 in table 4
- Row 210: Is the change of 25% significant due to high standard deviations? Add the statistical proof that the decreasing is really significant.
- Row 211: For evaluation of iodine content in boiling water the exact volume of the water is necessary. How was the volume of water after boiling measured? Were there some water losses? (residues on dishes or fillet, etc…).
General:
The name of Gadhus is used in whole text, but the real name is Gadus
Please distinguish word “concentration” and “content”, concentration is quantity related to volume (e.f. mg/L) but content is quantity related to mass (e.g. mg/g). These word are used wrongly in the manuscript.
The quality of the manuscript is not high and manuscript processing is not careful. It contains many formal and grammatical mistakes, and too many inconsistencies.
Author Response
Dear reviewer,
-thank you for many good comments. Our replies are in the attached file.
Lisbeth

Reviewer 2 Report
In this manuscript, authors compare various cooking methods of Atlantic cod to see if residual contents of iodine and mercury change before and after cooking. Overall, it is very practical and of interest to the readership of Foods. However, the observations are only empirical and scientific explanation is weak. Inorganic elements such as iodine and mercury are found in diverse species in organic matrices such as foods. The ICP-MS has limitations of providing the speciation of iodine and mercury and it is difficult to generalize the main idea of this paper and why certain methods of cooking significantly reduce the contents of iodine and mercury. Specific comments on the manuscript are as follows:
- Loss of iodine in thawing water is an important finding of the study. This loss could be thought as a dilution effect and it is reasonable to assume that the amount of thawing water to mass of cod should be an important factor determining the loss of iodine. However, specific information on the experimental conditions is not mentioned.
- In relation to comment 1, idodide ion is highly soluble in water. Is the loss of iodine by the dissolution of iodide or other anionic form of iodine? Without further investigation of speciation of iodine, we cannot provide a mechanistic understanding of cooking methods on the removal of iodine.
- Mercury concentration after cooking slightly increased as shown in Table 4. Is it simply due to the loss of total weight of cod? To compare mercury content before and after cooking, it is desired to normalize the content on dry-weight basis because mercury would not dissolve in water phase of cod.
Minor comments:
Line 124: standard deviation with n=2?
In Figure 2, please explain esterisks.
Author Response
Dear reviwer,
Thank you for your comments. Our replies are given in the attached file.
Lisbeth

Reviewer 3 Report
I applaud the work by the authors. However, there are some serious flaws/shortcomings with this article:
- There has been some relevant research omitted (other seafood but relevant). They show that cooking does not change mercury/metals (logical). Here are some: Morgan, J. N., Berry, M. R., & Graves, R. L. (1997). Effects of commonly used cooking practices on total mercury concentration in fish and their impact on exposure assessments. Journal of Exposure Analysis and Environmental Epidemiology, 7(1), 119. Perello, G., Marti-Cid, R., Llobet, J. M., & Domingo, J. L. (2008). Effects of various cooking processes on the concentrations of arsenic, cadmium, mercury, and lead in foods. Journal of agricultural and food chemistry, 56(23), 11262-11269.
- The experimental design is not there. All that is presented is data from duplicates or more (they do not mention how many subsamples). There are no replications and thus there is no way to find out variation
- I did not go any further since I do not consider the results valid from a statistical standpoint but they show what others have shown already
- The conclusions speak for themselves: "The present study has determined the effect of thawing and different cooking methods on the 297 content of iodine and mercury in Atlantic cod. Boiling decreased on average the iodine content in the 298 fillet with 25%. Pan-frying and oven-baking caused minor changes in the net iodine content, while 299 concentrations increased due to loss of water. For mercury content or concentration, we found 300 minimal changes of the different cooking methods. Further studies are warranted to better 301 understand the variation in iodine concentration in fish fillets and type of processing should be 302 specified in food composition databases. " Variation was not found due to lack of replications.
Author Response
Dear reviwer,
-thank you for your comments. Our replies are given in the attached file.
Lisbeth Dahl

Reviewer 4 Report
Presented manuscript is very interesting and is within the scope of the journal. The subject of the study is very important, since majority of the recent studies deals with calculations using databases and not with analytical determinations of various nutrients and contaminants. Therefore analytical data on the food composition are of great importance in relation to human health. Below are some minor errors, which need to be corrected or explained:
- Abstract, line 12 - it is better to write determined instead of "examined", as the latter one is mostly reffered to people
- line 35 - please put the number in bracket for "Erkan 2011". Lack in References.
- line 52 - "and considered relative narrowed compared to other micronutrients (NNR 2012)" - this sentence is not completely understandable to me. "Relative narrowed compared to other nutrients"? Please elaborate on this thought.
- Table 1 - please put range of results for each mean value, as the SD values are very high
- line 250 - "observed a minor increase of the moisture after pan-frying and oven-baking"? Increase of the moisture? Rather decrease, as the dry weight increased (Table 2)
Author Response
Dear reviwer,
thank you for your comments. Our replies are adressed in the attached file.
Lisbeth Dahl

Round 2
Reviewer 2 Report
In the revised manuscript, authors addressed most of concerns raised by the reviewer. It is publishable after a few minor spell-checks and minor editorial corrections.
Reviewer 4 Report
The manuscript was significantly improved. The Auhors referred to all my comments and questions and introduced the necessary changes to the manuscript. I do believe that the purpose of the study is interesting and is within the scope of the journal.
This manuscript is a resubmission of an earlier submission. The following is a list of the peer review reports and author responses from that submission.